# Identification of Amino Acids and Polyphenolic Metabolites in Human Plasma by UHPLC-ESI-QTOF-MS/MS, after the Chronic Intake of a Functional Meal in an Elderly Population

**DOI:** 10.3390/foods13162471

**Published:** 2024-08-06

**Authors:** Alma A. Vazquez-Flores, Óscar A. Muñoz-Bernal, Emilio Alvarez-Parrilla, Alejandra Rodriguez-Tadeo, Nina del Rocío Martínez-Ruiz, Laura A. de la Rosa

**Affiliations:** 1Departamento de Ciencias Químico-Biológicas, Instituto de Ciencias Biomédicas, Universidad Autónoma de Ciudad Juárez, Av. Benjamín Franklin No. 4650, Zona PRONAF, Ciudad Juárez 32315, Chihuahua, Mexico; alma.vazquez@uacj.mx (A.A.V.-F.); adrian.munoz@uacj.mx (Ó.A.M.-B.); nmartine@uacj.mx (N.d.R.M.-R.); 2Departamento de Ciencias de la Salud, Instituto de Ciencias Biomédicas, Universidad Autónoma de Ciudad Juárez, Av. Benjamín Franklin No. 4650, Zona PRONAF, Ciudad Juárez 32315, Chihuahua, Mexico; alrodrig@uacj.mx

**Keywords:** phenolic metabolites, aminoacidic metabolites, geriatric nutrition, multivariate statistical methods, *Brosimum alicastrum*

## Abstract

Novel foods especially formulated and targeted for the elderly population should provide sufficient nutrients and bioactive ingredients to counteract the natural age-related deterioration of various organs and tissues. Dietary protein and phenolic compounds achieve this goal; however, older adults have alterations in their gastrointestinal system that may impact their bioavailability and few studies have been aimed at this population. Since phenolic compounds are the subject of multiple biotransformations by host and microbiome enzymes during the digestion process, identification of their bioavailable forms in human plasma or tissues represents a considerable analytical challenge. In this study, UHPLC-ESI-QTOF/MS-MS, chemometrics, and multivariate statistical methods were used to identify the amino acids and phenolic compounds that were increased in the plasma of elderly adults after a 30-day intervention in which they had consumed an especially formulated muffin and beverage containing *Brosimum alicastrum* Sw. seed flour. A large interindividual variation was observed regarding the amino acids and phenolic metabolites identified in the plasma samples, before and after the intervention. Three phenolic metabolites were significantly increased in the population after the intervention: protocatechuic acid, 5-(methoxy-4′-hydroxyphenyl) valerolactone, and phloretic acid. These metabolites, as well as others that were not significantly increased (although they did increase in several individuals), are probably the product of the microbiota metabolism of the major phenolic compounds present in the *B. alicastrum* Sw. seed flour and other food ingredients. A significant decrease in 4-ethyl-phenol, a biomarker of stress, was observed in the samples. Results showed that the incorporation of foods rich in phenolic compounds into the regular diet of older adults contributes to the increase in bioactive compounds in plasma, that could substantially benefit their mental, cardiovascular, and digestive health.

## 1. Introduction

The population of elderly adults increases yearly in Western cultures; this comes with the challenge of meeting their specific health needs due to their bodies’ deterioration [1]. The prevalent musculoskeletal fragility, alterations in cardiovascular function, deterioration and dysfunction of the gastrointestinal system, and continuous neuronal degeneration are conditions that commonly compromise the quality of life of this population. Therefore, it is imperative to propose strategies that help prevent aging-related diseases [2]. At present, “nutrigerontology” is viewed as an efficient option to counteract various diseases associated with aging [3] through the incorporation of adequate amounts of proteins, vitamins, fatty acids, sterols, and bioactive compounds into the diet, promoting a good state of health in the elderly [4].

Adequate protein consumption in the diet is essential for maintaining endogenous proteins at an optimal level and for the correct functioning of tissues and organs [5,6,7]. Diseases of the skeletal-muscular, cardiovascular, and neuronal systems are associated with a protein deficiency in the elderly. Protein intake for the elderly is recommended in a range of 1.5 to 2.0 g of protein per kg of body weight daily [5]. In several countries, including Mexico, the habitual diet of older adults barely manages to cover 80% of these protein needs [7,8]. Hence, adequate protein consumption through dietary supplements or enriched foods seems to play an essential role in the health of older adults [5]. Likewise, incorporating foods rich in bioactive compounds, such as polyphenolic compounds, promotes the health status of consumers. For example, consumption of chocolate polyphenols reduced cardiovascular risk, blood pressure, and hypertension prevalence in older adults [9]. Furthermore, adding supplements formulated from blueberries improved neuronal and cognitive conditions in geriatric patients [10].

*Brosimum alicastrum* Sw. is a native tree species of the Mexican tropics whose seed possesses a high content of protein and polyphenolic antioxidants [11]. *B. alicastrum* seed has been widely consumed by the Mayan culture since 300 A.D., and in recent years has attracted interest due to its high nutritional profile [11]. A previous study showed that consumption of foods enriched with *B. alicastrum* Sw. seed flour improved the cardiovascular health of older adults. The authors reported that after 30 days of consuming a beverage and muffin containing *B. alicastrum* Sw. seed flour, blood glucose levels remained optimal and plasma LDL cholesterol decreased. Furthermore, consuming the beverage and muffin increased body weight, preserved muscle reserves through increased protein intake and metabolism, and improved the antioxidant activity in the plasma of older adults [8]. Thus, we hypothesized that the extra protein and phenolic compounds found in the *B. alicastrum* Sw.-enriched foods could be the main ingredients responsible for the food’s beneficial effects.

It is well known that dietary protein must be completely hydrolyzed by gastric and enteric peptidases in the digestive system, so their constituent amino acids may be absorbed into the bloodstream. Food polyphenolic compounds are poorly bioavailable in their native form; they are mostly conjugated by enteric and hepatic enzymes or biotransformed by enzymes from the microbiota. This releases diverse metabolites into the bloodstream that often exhibit a biological activity greater than their precursors present in the food [9,12]. The identification of metabolites derived from protein and especially polyphenol consumption in foods represents a considerable analytical challenge. In recent years, these studies have been addressed by the development of analyses of the complete metabolome of plasma, urine, or tissue samples that make use of chemometric multivariate statistic tools. Moreover, older adults have alterations in their gastrointestinal system that may impact nutrient bioavailability. The increased pH in the stomach decreases the degradability of the food matrix, the secretion of efficient digestive enzymes decreases, and the morphological changes of the enterocyte and microbial dysbiosis modify the release and use of metabolites in the older adult [13]. Thus, this study aimed to identify the metabolites present in the plasma of the elderly adults who had consumed the foods enriched with *B. alicastrum* Sw. seed flour, through UHPLC-ESI-QTOF/MS-MS, as evidence of the biotransformation and absorption of proteins and polyphenolic compounds in the geriatric population.

## 2. Materials and Methods

### 2.1. Plasma Samples

The bioassay to evaluate the effect of the consumption of two fortified foods during breakfast and dinner on the heath stats of the elderly was previously published [8]. Briefly, twenty elderly adults (>60 years; 12 men and 8 women), residents of three nursing homes in Ciudad Juárez, Chihuahua, were selected. According to the Geriatric Depression Scale and the Pfeiffer scale, all the individuals showed cognitive capacity and moderate anxiety-depressive state. Those individuals who presented a diagnosis or blood test with alterations in the liver and kidney were excluded. All participants agreed to the experimental procedures described in the project approved by the bioethical committee from Universidad Autónoma de Ciudad Juárez (UACJ) (CIBE-2017-1-47 and CIBE-2018-1-37). The foods enriched (muffin-type bread (51 g) and “atole”-type beverage (100 mL)) with *Brosimum alicastrum* Sw. seed flour were designed with a soft consistency, and a significant contribution of proteins (14.7 g per serving), dietary fiber, and polyphenolic compounds. The formulation of the foods and the proximate composition of each type of food were previously reported [8,14]. The study considered an initial control, with the usual diet of each residence, and an initial blood sample was taken. For the intervention, the participants received 51 g of muffin-type bread and 100 mL of “atole”-type beverage during breakfast and dinner for 30 consecutive days. At the end of this period, a second blood sample was taken (intervention). Both blood samples (control and intervention) were transported to the UACJ facilities and processed to separate the plasma from the packed blood cells. Plasma was separated by centrifugation at 3000 rpm for 15 min at 4 °C (Sorvall 16R, Thermo Scientific^®^, Waltham, MA, USA). The supernatant was recollected, transferred to microtubes, and stored at −80 °C until subsequent analysis.

### 2.2. Phenolic and Amino Acid Metabolites Extraction

Plasma samples from each participant (control and intervention) were subjected to an extraction of metabolites. The extraction was performed according to the previous methodology described by Muñoz-Bernal et al. [15]. In brief, 200 μL of plasma was mixed with 1000 μL of absolute ethanol (Molecular Biology, Merck^®^, St. Louis, MO, USA). The samples were centrifuged at 13,000 rpm for 5 min at room temperature. The supernatant was collected, and the pellet was mixed again with 1000 μL of absolute ethanol and centrifuged under the same conditions. Both supernatants were collected and dried under N_2_ gas. Then, the dry metabolites samples were resuspended with 100 μL of a mixture of acetonitrile:water (50:50).

### 2.3. Metabolite Analysis by Liquid Chromatography Coupled with Mass Spectrometry in Tandem

Resuspended metabolite samples were filtered through a 0.45 μm Nylon syringe filter (13 mm; Titan 3, Thermo Scientific^®^, Waltham, MA, USA) and placed in amber vials. Samples were analyzed in a 1290 Infinity series ultra-high performance liquid chromatography system (UHPLC) (Agilent Technologies, Inc., Santa Clara, CA, USA). The system consisted of a 1290 Infinity quaternary pump with a built-in degasser (set at 0.4 mL/min), a 1290 Infinity autosampler with temperature control (kept at 25 °C), a 1290 Infinity thermostated column compartment (maintained at 25 °C), and a 1290 Infinity diode array detector (set at 220, 320, and 370 nm). A ZORBAX C_18_ column (Rapid Resolution High Definition, 50 mm × 2.1 mm, 1.8 μm, Agilent Technologies, Inc., Santa Clara, CA, USA) was used for separation. Three μL of each sample was injected into the system. The separation of the metabolites was carried out through two mobile phases: (A) 0.1% formic acid (HPLC, Merck^®^, St. Louis, MO, USA) in water (LC-Mass Spec, Tedia^®^, Fairfield, OH, USA); and (B) acetonitrile (LC-Mass Spec, Tedia^®^, Fairfield, OH, USA) 100%. The following gradient was used: 0–1 min 90% A, 1–4 min 70% A, 4–6 min 62% A, 6–8 min 40% A, 8–8.5 min 40% A, 8.5–10 min 10% A. Each sample was injected in triplicate.

The mass spectrometer was an Agilent 6530 Accurate-Mass Q-TOF MS/MS equipped with electrospray ionization (ESI) (Agilent Technologies, Inc., Santa Clara, CA, USA). The mass spectrometer was used in negative mode, and nitrogen (N_2_) was used as a drying gas at 340 °C with a flow rate of 13 L/min. The capillary voltage was set at 4000 V and 60 psi, fragmentor voltage 175 V, and skimmer 65 V. The mass scanning was set in a mass-to-charge (*m*/*z*) ratio of 100–1100 for MS and 50–1000 for MS/MS.

Mass Hunter Qualitative software version B.07.00 (Agilent Technologies, Inc., Santa Clara, CA, USA) was used to identify the metabolites. The parameters to identify the metabolites were the exact mass of the compounds, the isotopic distribution, and the fragmentation pattern. The mass compound was compared with a specific database of phenolic compounds, phenolic metabolites, and amino acids [15]. All compounds dissimilar to the theoretical mass under 80% were discarded. The results for each metabolite were expressed in the abundance of the ion; the limit for identification was 3 times the signal-to-noise ratio (S/N) (S/N = 12,525).

### 2.4. Statistic Analysis

Aminoacidic and polyphenolic metabolites’ differences between control and intervention samples were analyzed through different multivariable (chemometric) analyses using XLSTAT for life sciences software (Addinsoft^®^, Paris, France) version 2024.1. A comparative heat map of polyphenolic metabolites was carried out in the control and intervention groups, using the differential abundances of the ion. A volcano plot was conducted to determine significant changes before and after consuming enriched foods. A *p*-value = 0.05 was used to determine significant differences, and a post hoc Tukey was performed.

## 3. Results and Discussion

### 3.1. Amino Acids in Plasma

One of the main challenges in the diet of older adults is preserving the quality of their nutrients; adequate consumption of fats, carbohydrates, and proteins is essential to maintaining the body′s quality of life [6]. So, developing foods that promote the absorption and increase in proteins in optimal quantities is imperative.

The spectral information of all amino acids identified in plasma samples is shown in Appendix A. The content of leucine (Leu, blue dot) increased more than twice in the plasma of geriatric patients, after consumption of the foods enriched with *Brosimum alicastrum* Sw. seed flour, as a protein-rich ingredient (29 g per day). Phenylalanine (Phe), tryptophan (Trp), and valine (Val) were modestly increased (gray dots), and tyrosine (Tyr, red dot) decreased almost two times, while minor decreases were observed for glutamate (Glu), lysine (Lys), and proline (Pro, Figure 1). Nevertheless, none of these changes was significant (*p* > 0.05). It has been demonstrated that older adults have a lower level of amino acids in plasma and a low ratio of essential to non-essential amino acids. This effect has been justified by the multiple digestive changes that older adults go through with aging [16].

It is important to mention that Leu, Phe, Val, and Trp are considered essential amino acids, and it has been observed that increasing these types of amino acids in the diet promotes muscle function in geriatric patients, even during periods of rest (patients in bed) [17]. Also, it has been reported that Leu, Val, and Phe are associated with increased longevity [17] since consuming these amino acids reduces the risk of muscle strength loss in the Chinese geriatric population [18]. The requirement of Leu in the geriatric population must increase above the recommendation for younger populations (34–39 mg/kg of body-weight/day), and double to adequately regulate muscular protein synthesis, prevailing adequate protection of bone tissue and increasing the quality of life of the older people [19].

The bioavailability of Trp as an essential amino acid also positively impacts consumers’ mental health since it is transformed by intestinal epithelial cells through the enzyme tryptophan-hydroxylase to serotonin and melatonin [18]. Serotonin is a potent neurotransmitter that regulates the organism’s decision-making, cognitive, emotional, and eating processes [18]. Meanwhile, melatonin is a neurotransmitter that optimizes mitochondrial function and regulates the circadian cycle, improving the condition of patients with depression and anxiety [18]. Thus, the consumption of foods enriched with *B. alicastrum* Sw. seed flour could regulate appetite and sleep schedules, and improve the mental health of older adults.

The type and quantity of amino acids in plasma depend on several intrinsic factors of the food: the food matrix, the manufacturing process, and the structure of the proteins [19]. Other factors must be considered, such as the age, sex, and specific pathologies of the gastrointestinal tract of the person who ingests them. In older adults, the enteric cells use about 50% of the amino acids from dietary proteins. The rest must be distributed through the plasma to cover the different metabolic requirements, often insufficient to exert muscular and neuronal benefits [20]. For this reason, increasing amino acid content in the plasma of older adult population is desirable and foods enriched with *B. alicastrum* Sw. can aid in this task.

### 3.2. Phenolic Metabolites

Inclusion of the muffin and beverage with *B. alicastrum* Sw. seed flour in the geriatric diet not only complemented the caloric and nutritional requirements (proteins, unsaturated fatty acids), and micronutrients (Fe, Zn, Folic acid) of the geriatric patients; it also allowed the incorporation of polyphenolic compounds into the diet [8]. Polyphenolic compounds have been shown to modulate various physiological processes such as the cellular oxidative-reductive state, enzymatic activity, cell proliferation, and cellular signal transduction pathways [12]. Nonetheless, most of the pharmacological activity of polyphenolic compounds has been demonstrated in cellular and animal studies, while few studies have been carried out by clinical trials [21]. Advances in clinical studies are required to complete the information on the large number of biotransformations that polyphenols undergo through enteric, hepatic, and microbiota metabolism, and scientifically support their use as clinical adjuvants [12,21,22].

In this study, UHPLC-QTOF-MS/MS analysis of plasma from older adults allowed the identification of 27 polyphenolic metabolites. The spectral information of polyphenols and polyphenolic metabolites can be found in Appendix A. The heatmap (Figure 2) shows the relative change in peak area of each compound in each participant, before and after the 30-day intervention. Around 48% of the metabolites were unchanged through the intervention (white boxes). The boxes in blue indicate an increase in the metabolites’ relative concentration at the end of the intervention, and the color intensity indicates the fold change in the metabolite abundance. Red boxes indicate metabolites that decreased at the end of the intervention. Considering this, the metabolites that increased their abundance at the end of the intervention represent 31% of all metabolites, while 21% showed a decrease (Figure 2); therefore, more than 70% of the metabolites identified in samples maintained and even increased in abundance at the end of the intervention. Metabolites such as benzoic acid (BA), carnosic acid (CARA), hippuric acid (HPA), vanillin (VAN), 3-hydroxyphenyl valeric acid (3HPVA), homovanillic acid (HOMVA), and phloretic acid (PHOA) showed a mild increase (approximately 1-fold) in most samples. However, the changes in all metabolites before and after the intervention were highly heterogeneous among the participants; for example, 3HPVA was mildly increased in 13 participants, decreased in five samples and was unchanged in two. More metabolites were decreased after the intervention in participants 8, 9, and 19 than in the other participants (Figure 2), also highlighting the interindividual variation.

Through the years, the gastrointestinal system reduces the production of saliva, digestive enzymes, and hydrochloric acid in the stomach. This induces an increase in the stomach pH and thus hinders the biodegradability of foods, which is why older adults cannot absorb many nutrients and bioactive compounds [4]. Nevertheless, the results of the present study, finding an increase in some amino acids and phenolic compounds in some elders, show that the *B. alicastrum* Sw.-enriched muffin and beverage were good sources of easily digestible protein and bioactive compounds for older adults.

Although the heatmap in Figure 2 allows for a comparative analysis of the changes in the phenolic metabolites the plasma of each participant along the intervention, it cannot show if the changes are statistically significant in the studied population. Therefore, Figure 3 shows a volcano plot with the 27 polyphenolic metabolites identified in the plasma samples. Metabolites that showed no change (less than 1-fold) between the beginning and the end of the intervention are shown in gray, inside the two central dotted lines [23]. Metabolites that decreased in abundance after the intervention are shown in red (non-significant) and those that increased, in blue (non-significant) or green (significant increase, *p* < 0.05). The metabolites that increased significantly were protocatechuic acid (PROA), 5-(methoxy-4′-hydroxyphenyl) valerolactone (5MHPV), and PHOA, increasing their abundance three times over their initial value. Non-significant increases were observed for equol (EQ), CARA, 3HPVA, BA, HOMVA, *m*-hydroxybenzoic acid (MHBA), HPA, VAN, hesperetin (HES), and 3,4-hydroxytoluene (34DHT). Results show that the only polyphenolic metabolite reduced after the intervention in the elderly population, was 4-ethylphenol (4EP). These results indicate that the consumption of *B. alicastrum* Sw.-enriched muffin and beverage increased a diversity of polyphenolic metabolites in plasma of older adults.

Previous works have identified phenolic compounds in *Brosimum alicastrum* Sw. seeds, most are linked to organic acids, sugars, and even other polyphenolic compounds (polymeric forms) [9,24,25]. Phenolic compounds are they are poorly bioavailable in these forms; thus, they are seldom identified in plasma. It is estimated that only 5 to 10% of ingested polyphenols are absorbed in the small intestine; they are then transformed (methylated, sulfonated, or glucuronidated) by enteric and hepatic phase II enzymes in the first 8 h after consumption, and, since they are recognized as xenobiotics, are quickly excreted in the urine [25]. Thus, a large proportion of polyphenols (80–90%) continue their way to the large intestine, where they are catabolized by the gut microbiota [25,26]. The colon and the gut microbiota function as a bioreactor that transforms unabsorbed polyphenols into smaller compounds (metabolites) through reactions of hydrolysis, dihydroxylation, demethylation, decarboxylation, and breaking of aromatic rings [9,12,25,27], releasing metabolites that are easier to absorb and present greater biological activity than their precursors in foods [22].

Since the plasma analysis was carried out 12 h after the last food intake, it was not possible to identify any phase II metabolites (oxidized, methylated, sulfonated, or glucuronidated polyphenols) [18,28]. Hence, the polyphenolic metabolites identified in plasma were products of the catalytic activity of the gut microbiota of the elderly individuals on the polyphenolic compounds previously reported in *B. alicastrum* Sw. [11] or other components of the muffin and beverage (Figure 4).

Among the compounds reported in the *B. alicastrum* Sw.-enriched foods used in the intervention, were chocolate procyanidins [14] and (epi)catechin gallate [11]. Procyanidins are a substrate for microbiota of the genera *Eubacterium* and *Eggerthella*, which manage to depolymerize these molecules and release flavan-3-ols, including (epi)catechin gallate [12]. These monomeric flavan-3-ols, in turn, are substrates for *Flavonifractor plautti* that allow the cleavage of the A ring, releasing dihydroxy phenyl valerolactones (Figure 4). The most common are 5-methoxy-hydroxyphenyl valerolactone (5MHPV) and 3,4-dihydroxy phenyl valeric acid (34DHPVA). Both molecules can be absorbed passively in the intestine, through the paracellular route, and increased significantly in plasma [12,29]. Valerolactone (5MHPV) increased significantly after consumption of *B. alicastrum* Sw.-enriched foods (Figure 3). These molecules have cardioprotective activity since it has been reported that they can reduce the adhesion of monocytes to the vascular endothelium, preventing the formation of atherosclerotic plaques [12]. Also, valerolactones can reduce systolic blood pressure due to their hypotensive effect and the inhibition of the angiotensin-converting enzyme [12,29].

Depending on the intestinal microbiota, once the A and C rings of the flavan-3-ols are degraded, they can also release phenolic acids (Figure 4), which are easily absorbed by the intestine [12]. Protocatechuic acid (PROA) is another of the metabolites that significantly increased its plasma levels (Figure 3) and is possibly associated with the flavan-3-ols present in the *B. alicastrum* Sw.-enriched foods. PROA is a molecule capable of reducing the neuronal inflammation process that stimulates the aggregation of amyloid brain proteins in older adults; such aggregation of proteins is associated with diseases like Parkinson′s and Alzheimer′s [30]. A study in animal models suggests PROA as an efficient neuroprotector, inhibiting inflammatory processes mediated by cyclooxygenase 2 (COX2) and interleukin 6 [21]. It is important to consider the beneficial effects of all the ingredients used to prepare the muffin-type bread, such as cocoa and almonds. Those ingredients are considered foods with a high content of proanthocyanidins, constituted of monomers of flavan-3-ols [31,32], which may contribute to the increase in PROA, 5MHPVA, and 34DHPVA in the plasma of older adults.

Phloretic acid (PHOA) was the third metabolite that increased significantly in plasma after consumption of the *B. alicastrum* Sw.-enriched foods (Figure 3). This metabolite can be the product of microbiota metabolism of the cinnamic acid (Figure 4) found in the muffin-type bread and “atole”-type beverage [11]. It has been reported that some lactic-fermenting bacteria (*Lactobacillus* and *Enterobacteriaceae*) use cinnamic acids in redox reactions as a reductant to produce extra ATP for their metabolism, releasing PHOA molecules [33]. Additionally, ferulic acid has been reported in *Brosimum alicastrum* Sw. seed flour as ferulic acid esterified with quinic acid [11]; some enterobacteria can uptake ferulic acid and subject it to demethylation and dihydroxylation, releasing PHOA. PHOA is a short chain fatty acid known as 3-(4-hydroxyphenyl) propionic acid. It is a biologically relevant metabolite as it has been observed to have antipyretic, anti-inflammatory, and analgesic effects in vitro [34].

Metabolites that increased at the end of the intervention, although not significantly in the population of older adults, include homovanillic acid (HOMVA) and 3,4-dihydroxy toluene (34DHT). Both are metabolites of the intestinal microbiota when rutin and quercetin are consumed [12] (Figure 4). Quercetin is one of the most abundant polyphenols in the plant kingdom and was identified in the *B. alicastrum* Sw. seed flour [11]. Isoquercetin is a glycosylated quercetin that once it enters the small intestine is deglycosylated by the enzyme lacto-phloretin hydrolase (β-hydrolase family) of the intestinal brush, releasing quercetin (aglycone) [35,36]. Quercetin in the large intestine can be a substrate to bacteria from the genera *Eubacterium*, *Clostridium*, *Bifidobacterium*, *Lactobacillus*, and *Streptococcus*, which fission the C ring and release molecules such as HOMVA and 34DHT [31,36]. These lower molecular weight phenolic acids can easily enter the bloodstream by the paracellular route [27] and increase its plasma content. It has been demonstrated that HOMVA and 34DHT can reach various tissues, decreasing the glycosylation of lysine from collagen and other proteins by up to 98 and 93.4%, respectively. The formation of glycosylated proteins is associated with their malfunction and can induce the development of several neuropathies, nephropathies, retinopathies, cardiovascular disorders, and Alzheimer′s disease [12].

The increase in vanillin (VAN) can be associated with glycosylated vanillic acid, previously reported in *B. alicastrum* Sw. seed flour [11]. It has been reported that VAN can be deglycosylated in the small intestine by cytosolic hydrolases (β-hydrolases) [26]. Like the other low molecular weight polyphenolic metabolites, VAN easily enters the bloodstream by passive diffusion [27]. VAN is a metabolite shown to have neuroprotective activity against Alzheimer’s because of its antioxidant role in neutralizing reactive oxygen species, but also as an inhibitor of acetylcholinesterase. High levels of acetylcholine are responsible for the cognitive loss experienced by Alzheimer’s patients, and VAN has demonstrated the inhibition of acetylcholinesterase with an IC_50_ of 84.6 μg/mL. Furthermore, VAN inhibits the degradation of typical dopaminergic neurons in patients with Parkinson′s through its antiapoptotic activity in preserving mitochondrial function [37].

Hippuric acid (HPA), *m*-hydroxybenzoic acid (MHBA), and benzoic acid (BA) are metabolites with a very similar structure that increased at the end of the intervention. These metabolites may derive from chlorogenic and dicaffeoyl quinic acids, identified in the *Brosimum alicastrum* Sw. seed flour [11]. Chlorogenic acid is one of the polyphenolic acids most distributed in the plant kingdom [38]. Once chlorogenic acid enters the large intestine, bacteria of the genus *Bifidobacterium* [39] metabolize it, releasing HPA (Figure 4). Nowadays, low concentrations of HPA are associated with some of the physiological deterioration of older adults: muscle wasting, altered muscle metabolism, and loss of muscle mass [40]. HPA can, in turn, be biotransformed into MHBA and BA (Figure 4). However, these metabolites can also be catabolic products of gut microbiota from (epi)catechin gallate [11,41]. MHBA and BA possess antibacterial and antiviral activities in vitro [42,43]; moreover, MHBA and BA participate as prebiotics in older adults, modulating and regulating microbiota dysbiosis [39].

On the other hand, 4-ethylphenol (4EP) was the only metabolite that decreased at the end of the intervention. 4EP is produced from coumaric acid, which is present in *B. alicastrum* Sw. seed flour as a quinic acid ester [11], and passes through the small intestine without further modification. A recent study indicates that coumaric acid is a substrate for bacterial enzymes from the genera *Clostridium* and *Bacteroides* in the gut microbiota. 4EP is a biologically active metabolite at the neuronal level, inducing stress and depression process in older adults [44]. In this study, 4EP decreased in the plasma of older adults, despite the presence of coumaric acid in the diet. This may be due to the prebiotic effect exerted by other metabolites, inhibiting the reproduction of bacteria of the genus *Clostridium* and *Bacteroides*, as has been shown by the ingestion of wine polyphenols [44]. These results suggest a probiotic effect of *B. alicastrum* Sw. seed flour, so further studies should be carried out in order to analyze this intriguing possibility.

A few polyphenolic metabolites in the plasma samples could not be associated with the *B. alicastrum* Sw. seed flour [equol (EQ), hesperetin (HES), and carnosic acid (CARA)], but rather with the diversity of ingredients used to prepare the foods and increase their palatability. The increase in EQ can be associated with daidzein and other isoflavones abundant in soy products, which are metabolized by bacteria from the genera *Lactococcus* (hydrogenation), and *Eggerthella* (dehydroxylation) of the small intestine [12]. EQ is a molecule that can affect estradiol receptors and helps to regulate hormonal changes in older women [45]. The HES found in plasma samples may derive from glycosylated flavonoids in the cocoa from the muffin-type bread [11]. Once flavonoids enter the large intestine, they can be deglycosylated through the activity of β-glucosidases (rhamnosidases) of *Bifidobacteria* [46], allowing their entry to the bloodstream in the form of HES [47]. HES has demonstrated antioxidant and cardiovascular benefits since it protects the endothelium from oxidizing agents and promotes nitric oxide synthesis (vasodilator), modulating systolic and diastolic pressure [46]. Finally, CARA consists of a polyphenolic terpene, which may derive from the metabolism of fatty acids present in the muffin-type bread. Biologically, it has anti-inflammatory and neuroprotective potential.

Considering these results, the incorporation of foods enriched with *B. alicastrum* Sw. seed flour into the regular diet of older adults contributes to the increase in bioactive compounds in plasma, that could substantially benefit their mental, cardiovascular, and digestive health.

## 4. Conclusions

The obtained results suggest that the incorporation of foods enriched with *Brosimum alicastrum* Sw. seed flour in the diet of older adults increases the content of metabolites derived from the digestion of protein and polyphenolic compounds. In this way, there are potential benefits of incorporating *Brosimum alicastrum* Sw. seed flour into daily nutrition, so enriched meals play a crucial role in enhancing dietary quality and promoting overall health in older adults. The increase in the essential amino acid Leu in plasma indicates a possible protective effect on muscular and neuronal alteration typical of older adults. The increase in polyphenolic metabolites in plasma, such as valerolactones and phenolic acids, is attributed to the digestive activity of gut microbiota, releasing smaller molecules that could freely cross the intestinal epithelium and positively impact the cardiovascular, mental, and digestive health. Together, these results provide an approach to understanding how foods enriched with functional ingredients are transformed and used by the body, as a promising strategy to preserve the life quality of older adults. To this end, more studies are recommended, to evaluate the clinical efficacy of these metabolites through monitoring changes in gut microbiota, cardiovascular and neuronal biomarkers, and cognitive tests, among others.

## Figures and Tables

**Figure 1 foods-13-02471-f001:**
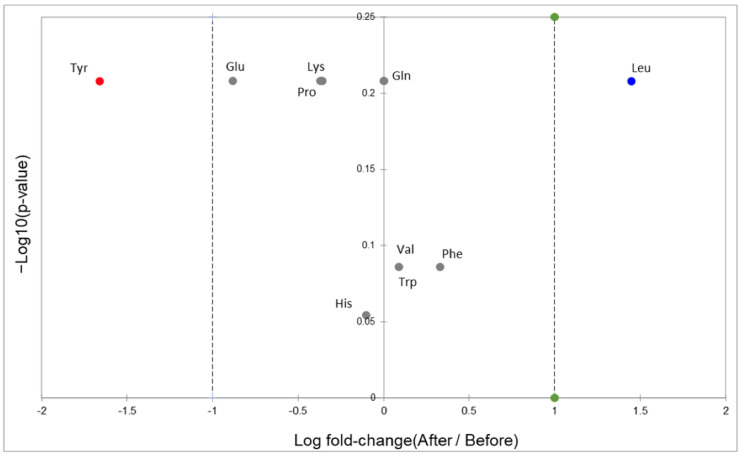
Volcano plot of amino acids found in older adult plasma samples after and before consuming foods enriched with *Borsimum alicastrum* Sw. seed flour.

**Figure 2 foods-13-02471-f002:**
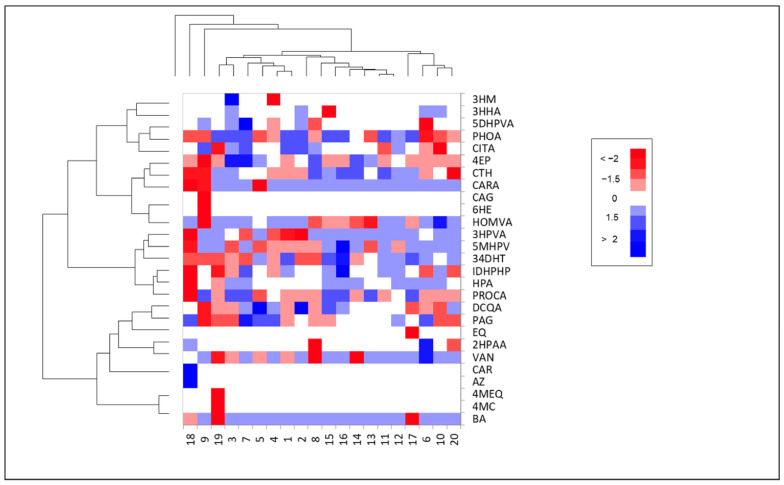
Heat map of polyphenolic metabolites at the end of intervention with foods enriched with *Brosimum alicastrum* Sw. seed flour. 3′-hydroxymelanettin (MH), 3-hydroxyhippuric acids (3HHA), 5-(3′,4′-dihydroxyphenyl)-valeric acid (5DHPVA), phloretic acid (PHOA), citric acid (CITA), 4′-ethylphenol (4EP), catechol (CTH), carnosic acid (CARA), caryatin glucoside (CAG), 6′-hydroxyenterolactone (6HE), homovanillic acid (HOMVA), 3-hydroxyphenyl valeric acid (3HPVA), 5-(3′-methoxy-4′-hydroxyphenyl)-valerolactone (5MHPV), 3,4-dihydroxytoluene (34DHT), isopropyl-3-(3,4-dihydroxyphenyl)-2-hydroxy-propanoate (IDHPHP), hippuric acid (HPA), Protocatechuic acid (PROCA), dicaffeoyl quinic acid (DCQA), phenylacetylglycine (PAG), equol (EQ), 2-hydroxyphenylacetic acid (2HPAA), vanillin (VAN), carnosol (CAR), azaleatin (AZ), 4′-methyl equol (4MEQ), 4′-methylcatechin (4MC), benzoic acid (BA).

**Figure 3 foods-13-02471-f003:**
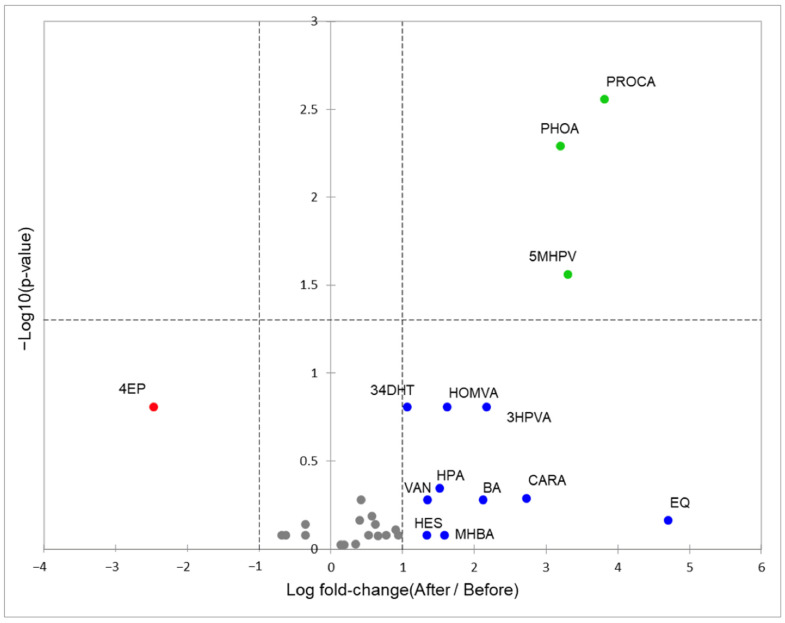
Volcano plot of differential abundance of polyphenolic metabolites before and after the consumption. Green dots express a significant difference at *p* < 0.05. Gray dots express polyphenolic metabolites without change during the intervention.

**Figure 4 foods-13-02471-f004:**
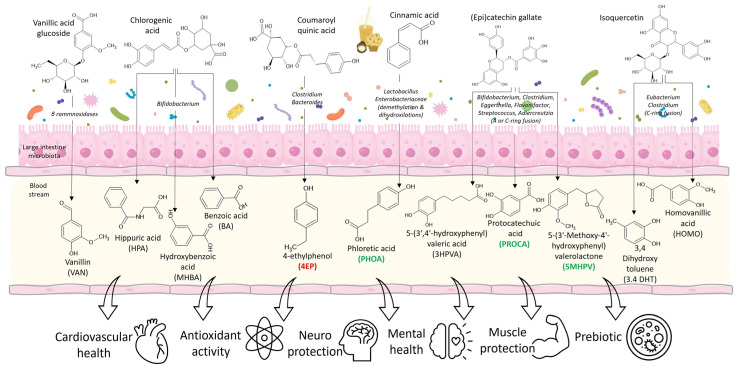
Polyphenolic compounds present in the foods enriched with *Brosimum alicastrum* Sw. seed flour and the possible biotransformation in polyphenolic metabolites by the gut microbiota.

## Data Availability

The original contributions presented in the study are included in the article and Appendix A, further inquiries can be directed to the corresponding author.

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
