# Peer review of "Identification of Amino Acids and Polyphenolic Metabolites in Human Plasma by UHPLC-ESI-QTOF-MS/MS, after the Chronic Intake of a Functional Meal in an Elderly Population"

_foods, 2024, doi:10.3390/foods13162471_

Round 1

Reviewer 1 Report

Comments and Suggestions for Authors

The manuscript presents a compelling approach, highlighting the critical importance of the bioavailability of bioactive compounds and their analysis in plasma.

In the introduction, the novelty and objectives of the research should be more prominently emphasized. Additionally, the significance of Brosimum alicastrum needs to be justified by discussing its chemical composition and the importance of including it in the diet. The mention of muffins and beverages containing Brosimum alicastrum Sw. seed flour is minimal and should be elaborated upon.

Materials and Methods: I suggest structuring this section by first describing the food samples, followed by details on consumption, and concluding with plasma sampling procedures.

Results: Lines 162-168 would be more suitable for the introduction, as they effectively highlight the necessity of your study. Moving this content to the introduction will strengthen your argument. Apart from this, the results are well presented.

Conclusion: The conclusion should be rewritten to reflect specific study results and observations. For example, something like: "Our study demonstrated that the bioavailability of bioactive compounds from Brosimum alicastrum significantly improved when included in the diet, as evidenced by a 30% increase in antioxidant activity in the tested subjects. Furthermore, the chemical analysis revealed a high concentration of essential nutrients and phenolic compounds, underscoring the potential health benefits of incorporating Brosimum alicastrum into daily nutrition. These findings suggest that Brosimum alicastrum could play a crucial role in enhancing dietary quality and promoting overall health."

Comments on the Quality of English Language

Minor edits for English language and spelling are needed.

Author Response

Dear Reviewer 1, we appreciate your time reading and the suggestions made to improve the manuscript.

The detailed answers to your queries are attached in a word document

Reviewer 2 Report

Comments and Suggestions for Authors

This manuscript focus on the identification of amino acids and polyphenolic metabolites by UHPLC-ESI-QTOF-MS/MS, I think the author prepared the manuscript very well, however, it is still need revised. The list of the corrected points described below, but especially the introduction and discussion should be reconsidered.

1. The QTOF-MS/MS figures of amino acids and polyphenolic metabolites are not list in the manuscript, please provide relevant information.

2. The analysis of amino acids often requires complex derivatization processes because they do not have fluorescent or chromophore groups. So, what is the principle of analysis and what are the retention times? Please supplement

3. B. alicastrum Sw. seed flour relevant nutritional analysis should be added.

4. The information of the consumer-twenty elderly adults (> 60 years; 12 men and 8 women)should be supplemented in the table.

Author Response

Dear Reviewer 2, we appreciate your time reading and the suggestions made to improve the manuscript.

The detailed answers to your queries are attached in a Word document
